# Development of a Rapid Gold Nanoparticle Immunochromatographic Strip Based on the Nanobody for Detecting 2,4-DichloRophenoxyacetic Acid

**DOI:** 10.3390/bios12020084

**Published:** 2022-01-30

**Authors:** Hui Zhou, Cong He, Zhenfeng Li, Jingqian Huo, Yu Xue, Xiaotong Xu, Meng Qi, Lai Chen, Bruce D. Hammock, Jinlin Zhang

**Affiliations:** 1College of Plant Protection, Hebei Agricultural University, Baoding 071001, China; zhouhui970618@163.com (H.Z.); hecong9696@163.com (C.H.); xy1135723614@163.com (Y.X.); xxt20011127@163.com (X.X.); menqi@ucdavis.edu (M.Q.); chenlai@hebau.edu.cn (L.C.); 2Department of Entomology and Nematology, and UCD Comprehensive Cancer Center, University of California, Davis, CA 95616, USA; lizhenfeng306@126.com (Z.L.); bdhammock@ucdavis.edu (B.D.H.)

**Keywords:** gold nanoparticle immunochromatographic strip, 2,4-Dichlorophenoxyacetic acid, nanobody, colorimetric card

## Abstract

2,4-Dichlorophenoxyacetic acid (2,4-D) is a systemic conductive herbicide widely used across the world. With the large-scale and continuous use of 2,4-D, its possible harm to the environment and non-target organisms has attracted increasing attention, and the construction of a stable rapid on-site detection method is particularly important. In order to achieve on-site rapid detection of 2,4-D, we developed a gold nanoparticle immunochromatographic strip method with the visual elimination value was 50 ng/mL, and a quantitative detection limit of 11 ng/mL based on a nanobody. By combing with the color snap, the immunochromatographic strip could quantitatively analyze the amounts of 2,4-D. Meanwhile, a colorimetric card based on the true color of the test strips was developed for the qualitative analysis of 2,4-D on-site. The samples (water, fruits and vegetables) with and without 2,4-D were detected by the immunochromatographic strips, and the results showed the accuracy and reliability. Thus, this assay is a rapid and simple on-site analytical tool to detect and quantify 2,4-D levels in environmental samples, and the analytical results can be obtained in about ten minutes. In addition, the nanobody technology used in this study provides an inexhaustible supply of a relatively stable antibodies that can be archived as a nanobody, plasmid or even its sequence.

## 1. Introduction

2,4-D, a systemic herbicide, is mainly used worldwide to control annual or perennial broadleaf weeds. Studies have found that long-term high-dose exposure to 2,4-D may cause some health problems in animals such as humans and rats [1,2], and their nervous systems will also be affected [3]. The extensive use of 2,4-D not only causes harm to the environment and non-target organisms, but also seriously affects the reproduction of aquatic and terrestrial organisms [4]. In the field, 2,4-D can be easily decomposed into 2,4-dichlorophenol, which is a toxic metabolite and can remain in the soil for a long time [5]. Recently, studies have shown that 2,4-D has strong mobility in urban soil, which may affect the quality of drinking water and the safety of aquatic organisms [6]. According to the World Health Organization standard (2011), the lowest residue limit of 2,4-D in drinking water is 30 ng/mL.

Nowadays, methods based on large-scale instruments for detecting 2,4-D can be roughly divided into three types: gas chromatography–mass spectrometry (GC-MS) [7], high-performance liquid chromatography (HPLC) [8,9], high-performance liquid chromatography/mass spectrometry (HPLC/MS) [10], and so on. For example, as early as 1996, Hong et al. detected the residues of 2,4-D in soil by gas chromatography with electron capture detection (GC-EDC) [11]. Cai et al. proposed a hypothetical principle of 2,4-D degradation using GC/MS, HPLC, ion chromatography, and other methods [12]. Goulart et al. developed a method to detect 2,4-D and its metabolites in water samples by solid-phase extraction (SPE) with liquid chromatography tandem mass spectrometry (LC-MS/MS) [13]. In addition to large-scale instrument detection methods, a variety of novel detection methods for 2,4-D have emerged in recent years. For instance, Zhang et al. developed [14] a ratiometric fluorescence method based on molecular imprinting, and the detection limit (LOD) was 90 nM (20 ng/mL). Xu et al. developed a nanogold surface-enhanced raman spectroscopy method for the detection of 2,4-D in environmental and food samples with an LOD of 0.79 pg/mL [15]. In summary, the instrument methods are accurate in results, but the preliminary sample preparation steps are cumbersome, and the detection requires expensive instruments and the extensive use of organic solvents. The chromatography based on instrument methods are of course sequential, while many sensors based on methods can be massively parallel. Moreover, the above detection methods have certain limitations in on-site rapid detection.

At present, enzyme-linked immunosorbent assay (ELISA) technology is increasingly used in the field of pesticide detection and analysis. Compared with instrument detection methods, the advantages of ELISA methods are more rapid, efficient and convenient [16]. Currently, according to cited references, ELISA is used to detect 2,4-D mainly based on polyclonal and monoclonal antibodies [17,18]. However, most of the conventional antibodies developed to date yield high cross-reactivities, because their immunogens are produced by the direct coupling of the carboxyl group in the 2,4-D structure to the protein [19,20]. This covers the most characteristic recognition site or distinguishing feature for 2,4-D. Therefore, their specificity is relatively poor. In addition, traditional antibodies have certain defects in stability. However, the high stability and high specificity that nanobodies can usually obtain [21] can well solve the problem of cross-reactivity, and are an important basis for constructing a rapid detection method on site.

Nowadays, because of the high stability and high absorption coefficient of gold nanoparticles (GNPs) [22], the nanogold immunochromatography strip with the advantages of portability, low cost, simple operation, and on-site rapid detection is widely used for detecting pesticides [23,24] in the environment. The gold nanoparticle immunochromatographic strips based on the nanobody may become an important basis for the rapid detection of pesticides on site. However, there are currently not many reports on the combination of nanobodies and GNPs.

In this study, a competitive immunochromatographic test strip was prepared based on a nanobody and gold nanoparticle, and could quantitatively analyze the 2,4-D in the environment sample. The test line was labelled with coating antigen and the control line was immobilized with anti-6 × His-tag antibody. In order to facilitate on-site rapid detection, we developed a “colorimetric card” based on the true color of the test strips, which was convenient for observing the content range of 2,4-D in samples on-site. The color snap APP was a very simple and practical screen color picker, and we could easily download it on Android or iPhone. It could be used to capture and identify a specific range of colors. In this work, we obtained a quantitative detection limit of 11 ng/mL by color snap APP. Therefore, the method developed in this study was of great benefit for on-site, fast and non-technical detecting.

## 2. Materials and Methods

### 2.1. Chemicals and Reagents

Nitrocellulose membrane, PVC pad, conjugate pad, absorbent pad, and sample pad were purchased from Shanghai Hualan Chemical Technology Company. His-Pur Ni-NTA columns and the rainbow 245 broad spectrum protein marker (11-245 KD) were purchased from Beijing Solarbio Co., Ltd. BSA protein, standard products (2,4-D, 2,4-dichlorophenoxybutyric acid, 2-methyl-4-chlorophenoxyacetic acid, 2-(4-chloro-2-methylphenoxy) propanoic acid, 2,4-D methyl ester, 2,4-D butyl ester, 2,4-dichlorophenol) were purchased from Thermo Fisher Scientific or Sigma-Aldrich (Shanghai, China). Anti-6 × His-tag antibody was purchased from Abcam. 2,4-D nanobody and coating antigens were kept in our laboratory. All reagents and solvents are of analytical grade.

### 2.2. Construction of Nanobody-Gold Nanoparticle

The GNPs were prepared according to the previous reports [23,25]. Different amounts of trisodium citrate were added to obtain two sizes of GNPs. Finally, the sizes of different gold nanoparticles were determined by transmission electron microscopy (TEM, Nippon Electronics Co., Ltd. JEM-1400, Tokyo, Japan).

The strain E. coli BL21 (DE3) was used to express the nanobody of 2,4-D. After the selection of induction time, induction temperature and isopropyl-β-D-thiogalactopyranoside (IPTG) concentration, the nanobody was expressed and purified by His-Pur Ni-NTA columns.

Before coupling, the optimal pH for the gold nanoparticle suspension was selected. At first, 100 μL gold nanoparticle suspensions were added to a 1.5 mL tube, and then different amounts of 0.1 M K_2_CO_3_ solution were added. After gentle mixing, 3 μL purified nanobody solution (at the concentration of 0.20 mg/mL) was added. Then, 10 μL 10% NaCl solution was added in after the suspension was left for 1 h at room temperature. Then, the suspension was mixed and left at room temperature for 1 h. Finally, the mixed solution was added into a 96-well plate, and the absorbance was measured at 520 nm by UV spectrophotometer. We selected the addition amount of K_2_CO_3_ and the optimum pH was obtained. At this pH, the absorbance was the highest. Similarly, according to the optimum pH, we selected the optimal amount of nanobody.

Based on the above results, 1 mL gold nanoparticle suspension and 0.1 M K_2_CO_3_ solution were added in the tube with stirring, and then nanobody was added and stirring continued for 30 min, followed by adding 80 μL 10% BSA and stirring for 20 min. After finishing, the combination was placed at 4 °C for 15 min, and then was centrifuged at 4 °C, 380× *g* for 15 min to remove the unconjugated nanogold particles and the supernatant was taken for another centrifugal to obtain the nanobody-gold nanoparticle pellet. After removing the supernatant, 1 mL gold nanoparticle resuspension was added slowly into the precipitation. Finally, the solution was centrifuged at 4 °C, 9500 *g* for 30 min; then, the nanobody-gold pellets were resuspended by 50 μL diluent, and stored at 4 °C until use.

### 2.3. Optimization of the Combination of Antigen and Nanobody for Immunochromatographic Strip

Immunochromatographic strips combine traditional immunolabeling technology with chromatography technology, and are based on formats in which free and immobilized antigen compete for binding to nanobody-AuNPs conjugates. As shown in Figure 1, the strips were assembled as follows: the nitrocellulose membrane was pasted on the center of PVC pad, and the absorbent pad was placed on the top of it, while the sample pad and conjugate pad were stacked on the bottom, in turn (Figure 1). Then, the test strips were cut with a size of 4 mm wide by an automatic cutter.

The 2,4-D coating antigen (at the concentration of 3.0 mg/mL) was spotted on the nitrocellulose membrane for test line at a dilution concentration of 1:1, 1:2, 1:4, 1:6, 1:8 through the coating buffer, and the nanobody (at the concentration of 0.20 mg/mL) conjugated with gold nanoparticle was spotted on the conjugate pad at a dilution concentration of 1:1, 1:2, 1:4, 1:6, 1:8 1:10 1:20 through the diluent. The control line was drawn with 0.5 µL anti-6 × His-tag antibody (at the concentration of 1.0 mg/mL). After placing at 37 °C for 1 h, 100 μL 0.01 M PBS (pH 7.4) containing 0.4% Tween 20 was added into the sample pad and the color was observed after 5 min.

### 2.4. Quantitative Detection of 2,4-D Based on the Color Snap APP

In this study, the color snap APP [23] was used for the quantitative detection of 2,4-D, and every test strip was measured 3 times. In short, one takes a photograph of the test strip under adequate lighting, and places the control line and test line in the center of the phone screen, then deletes the previous recognized colors in the software, and clicks “+” to select the control line and test line, finally clicking on the color name. According to the formulation [26]: Gray = R (Red) × 0.3 + G (Green) × 0.59 + B (Blue) × 0.11, the RGB of the control line and test line are calculated. In order to make on-site detection more convenient, we set a series concentration of 2,4-D standards (0, 1, 2, 3, 4, 5, 10, 20, 30, 40, 50 ng/mL) and a standard curve in which test/control (T/C) as the ordinate and 2,4-D concentration as the abscissa was obtained. In addition, we successfully made a “colorimetric card” based on the RGB values of the control line and test line obtained from the test strip. At last, the limit of detection (LOD) was calculated by ICH (the International Council for Harmonisation) guideline criteria as 3.3 σ/slope, where σ was the standard deviation of the blank treatments (*n* = 5) [27].

### 2.5. Cross-Reactivity

In order to evaluate the specificity of the immunochromatographic strip, the cross-reactivity (CR) of anti-2,4-D nanobody with structural analogues were also determined. Four compounds with similar chemical structures to 2,4-D were selected (Table 1). The relative CR was calculated by the following formula: CR (%) = [IC_50_ (2,4-D)/IC_50_ (tested compound)] × 100.

### 2.6. Stability of the Immunochromatographic Strip

The immunochromatographic strips were added with anti-6 × His-tag antibody and coating-antigen; then, they were stored at 4 °C, 25 °C and 37 °C for 1 d, 3 d, 5 d and 7 d. After different storage days, the analyte was applied separately to evaluate their stability. Three different concentrations of 2,4-D standard (0, 10, 100 ng/mL) were measured using the lateral-flow immunoassay.

### 2.7. Matrix Effect and Sample Analysis

In this study, the river water, grapes, tomatoes, cabbage and corn were selected for matrix effect evaluation. The river water was collected from Qingshui River, Nandi Road, Baoding City, Hebei Province and grapes, tomatoes, cabbage and corn were bought from the supermarket. The vegetable and fruit samples were frozen in liquid nitrogen, ground, and then 2 mL of methanol-containing PBS was added to 1 g of the samples, respectively. After vortexing, the mixture was centrifuged at 1500× *g* for 15 min, and the supernatants were collected and diluted 0, 2, 4, 8, and 16 times with 20 mM PBS to evaluate the matrix effect. The negative samples confirmed to be free of 2,4-D by LC-MS were spiked with 2,4-D at concentrations of 0, 1, 5, 10, 50, 100 ng/mL for recovery analysis.

## 3. Results and Discussion

### 3.1. Characterization of Gold Nanoparticles

In order to evaluate whether the size of the GNPs will affect the sensitivity of the gold nanoparticle immunochromatographic strip or not, in this study, two different sizes of GNPs with diameters of approximately 20 nm and 40 nm were synthesized and verified with a transmission electron microscope (TEM). Compared to the diameter of 40 nm, the GNPs with the diameter of 20 nm were more uniform in dispersion, and when adding the same amount of K_2_CO_3_ the color was brighter, and the absorbance was higher. So, we selected 20 nm of GNPs for the following study (Figure 2).

### 3.2. Construction of Nanobody-Gold Nanoparticle

The 2,4-D nanobody was obtained and expressed in our cooperative laboratory [21] through immunization with llama. The nanobody used for this study was obtained using the immunization antigen A, which was designed to retain the carboxyl functional group in the 2,4-D structure, because the carboxyl functional group was a most characteristic recognition site or distinguishing feature for 2,4-D. The structures of hapten A, hapten C, immunization antigen A, coating antigen C (Appendix A) used for this study, as well as the coupling method between haptens and protein, are shown in the Appendix A. The nanobody with the size about 15 kDa (Appendix A) was purified by His-Pur Ni-NTA columns and verified by SDS-PAGE. The optimal pH for the conjugation between nanobody and GNPs was determined by the addition of different amounts of 0.1 M K_2_CO_3_. The successful binding of GNPs to nanobody depended on pH. Generally, only when the pH was equal to the isoelectric point (PI) of the protein and was weakly alkaline, the two could be firmly bound. The change in absorbance was first increased and then decreased, only the pH corresponding to the maximum absorbance was equal to the protein isoelectric point (PI), which was slightly alkaline. With the increase in K_2_CO_3_, the absorbance at 520 nm had a tendency to increase first and then decrease (Appendix A). When the addition of K_2_CO_3_ was 5 μL, the absorbance under 520 nm reached the maximum, so, 5 μL of 0.1 M K_2_CO_3_ was selected as the optimum addition, and at this time the pH was 9.5. The method for determining the optimum amount of nanobody was based on the same method. With gradually increased amounts of nanobody at the optimum pH, the absorbance at 520 nm was the highest and the optimal conjugation state was reached when the amount of nanobody was 10 μL (at the concentration of 0.20 mg/mL). Because the amount of nanobody in the final coupling should be increased by 10%, the final amount of nanobody for conjugation was 11 μL (Appendix A). Based on these parameters, for 1 mL GNPs solution, the optimum conditions were 5 μL K_2_CO_3_ (0.1 mol/L) for pH adjustment and 11 μL nanobodies for conjugation.

### 3.3. Preparation and Optimization of Immunochromatographic Strip

In order to maximize the sensitivity of the immunochromatographic test strip, the checkerboard method was carried out to screen the concentration of nanobody and the coating antigen. The dilution times of the coating antigen and nanobody are shown in Appendix A. It can be easily observed that the color of the gold nanoparticle became significantly darker with the increase in coating antigen and nanobody, at last, four combinations with large dilutions and bright colors were finally selected, and the dilution times were 1:10, 1:20, 2:10 and 4:4 (nanobody: coating antigen), respectively. According to the four combinations selected above, the immunochromatographic test strip was tested with different concentrations of 2,4-D diluted with 0.01 M PBS. The results showed that in combination of 1:20 (nanobody dilution time: coating antigen dilution time), the test line color gradually became lighter with the increase in 2,4-D concentration, and the color almost disappeared at 50 ng/mL. In this combination, the visual detection limit was the lowest, so nanobody dilution time: coating antigen dilution time = 1:20 was determined as the final combination.

### 3.4. Sensitivity of Immunochromatographic Strip

Previous studies have reported that researchers constructed immunochromatographic test strips for the detection of 2,4-D based on traditional antibodies and up conversion nanoparticles (UCNPs) materials, with a detection limit of 5 ng/mL [28,29]. However, in the reported papers, traditional antibodies have certain limitations in terms of stability, so their practical application effect will be limited. In this study, the stable nanobody was used to replace the traditional antibody, which expands the scope for the application of test strips.

According to the optimal combination of coating antigen and nanobody, the test strips were used to detect different concentrations of 2,4-D diluted with 0.01 M PBS. It could be observed that the color of the test line was weakened with the increase in 2,4-D concentration, and the color disappeared at the concentration of 50 ng/mL, while the control line remained the same with the blank control (Figure 3a).

Based on the above concentrations of 2,4-D, the simulated strips and “colorimetric card” were made, so the changes of strips could be clearly observed (Figure 3b). The RGB of the control line and test line were obtained by color snap APP, and T/C was selected to be the *y*-axis of the curve. The value of T/C was a relatively stable parameter in calculation, and could effectively reduce the influence of the background [30]. With the increasing concentration of 2,4-D, T/C increased linearly, and the correlation was well between 0–50 ng/mL (R^2^ = 0.9979). We calculated the detection limit according to the ICH guideline, which was 11 ng/mL (Figure 3c).

It was evident that as the concentration of the analyte increased, the color of the test line gradually became lighter and the RGB became higher. When the concentration of 2,4-D reached 5 ng/mL, the test line became lighter. However, the test line disappeared completely and its RGB value was the highest at 50 ng/mL. Because of the difficulty of quantitative on-site detection, we prepared the test strip described here. The gold nanoparticle immunochromatographic strip method developed in this study could not only quickly quantify the samples, but could also estimate the 2,4-D content in the sample based on the colorimetric card.

### 3.5. Specificity of Immunochromatographic Strip

The immunochromatographic strip was also applied for the detection of 2,4-D structural analogues to evaluate the specificity. The concentration of all compounds used for the cross reactivity was 50 ng/mL. As shown in Table 1, except for 2-methyl-4-chlorophenoxyacetic acid, the test line colors of the other three compounds had no significant differences when compared with the blank control. The colors of the test line between 2-methyl-4-chlorophenoxyacetic acid and 2,4-D were close, because the structures of 2-methyl-4-chlorophenoxyacetic acid and 2,4-D were almost the same with each other, and Cl atoms were replaced with CH_3_. The cross-reactivity results showed that the test strip had good specificity among the compounds tested, which was consistent with the results obtained by the ELISA method [21]. In this study, the immunization antigen and coating antigen were heterologous, which was crucial in improving the sensitivity and specificity of nanobody.

### 3.6. Stability of Immunochromatographic Strip

The stability of immunochromatographic test strips have some economic values, so the storage time of test strips has attracted much attention. All test strips for stability tests were of the same batches, and three concentrations of 2,4-D (0, 10, and 50 ng/mL) were selected for the stability test. The strips were stored at different temperatures for 1 d, 3 d, 5 d, 7 d (Figure 4). The results showed that all the test line of strips at 50 ng/mL were relatively stable after 1 d. As the storage time of the strips increased, their stability also changed at different temperatures. The results showed that strips held at 37 °C showed instability evident after 3 days. Similarly, after 5 days, the stability at 25 °C decreased. After 7 days, only the strips stored at 4 °C were almost unchanged compared to the previous ones. One of the possible reasons was that higher temperature may significantly contribute to the denaturation of protein components in the strips; however, lower temperature at 4 °C could greatly reduce the influence of temperature.

In this study, the visual elimination value of the strips was 50 ng/mL, which was far below the national residue standards (GB2763-2019) in China. At the same time, when used with color snap APP, the detection limit of the strips was 11 ng/mL. From the perspective of stability, it could be kept at 4 °C for at least 7 d, which was quite better for on-site use and commercial storage. In the future, the test strips could be hold in a humid chamber and in a desiccated chamber to see which work better, and which could be used to evaluate the effect of humidity on the stability of the test strips.

### 3.7. Analysis of the Spiked Samples

The matrix effect is common in ELISA and can often significantly interfere with the analysis process of the analyte, and affect the accuracy of the results. In order to minimize the influences of matrix on the detection, the samples were often diluted with 0.01 M PBS buffer. Since the real sample often needs to be extracted with organic solvent-containing extracts, it is necessary to evaluate the influence of organic solvents on the test strip. Here, we selected methanol as the organic solvent. Then, 40%, 20%, 10% and 5% methanol/PBS containing 50 ng/mL 2,4-D were used as the detecting solvent to evaluate the effect of organic solvents on the test strip. The results showed that 5% methanol/PBS had almost no effect on the test results (Figure 5).

In addition to the direct dilution of the river water samples, the grape, tomato, cabbage and corn samples were extracted with 5% methanol/PBS, and the supernatant was diluted 0, 2, 4, 8, and 16 times to evaluate the matrix effect. The results (Figure 6) showed that the grapes and tomatoes matrix had no effect on the test strip when the supernatant was diluted by 2 times, the cabbage matrix had no effect on the test strip when diluted by 4 times, and corn matrix had no effect on the test strip when diluted by 8 times. The river water matrix had no effect on the test strip when diluted by 4 times.

Different concentrations 2,4-D were added to the river water, grapes, tomatoes, cabbage and corn samples that were confirmed to be free of 2,4-D through LC-MS detection. The results (Figure 7) showed that the detection line almost completely disappeared when the samples of river water, grapes, tomatoes, cabbage and corn were added with a final concentration of 50 ng/mL 2,4-D, which was consistent with the blank control.

## 4. Conclusions

This work successfully developed an immunochromatographic strip based on a nanobody to detect 2,4-D in the environment. By combing with the color snap APP, the immunochromatographic strip could quantitatively analyze the amounts of 2,4-D. At the same time, a colorimetric card was developed for the qualitative analysis of 2,4-D on-site, which had the advantage of being fast, convenient and sensitive. The immunochromatographic strip had a visual elimination value of 50 ng/mL and a quantitative detection limit of 11 ng/mL. It is sensitive enough to analyze 2,4-D in water, fruit, and vegetable samples. The method developed in this study has an important reference value for the rapid detection of 2,4-D in the environment.

## Figures and Tables

**Figure 1 biosensors-12-00084-f001:**
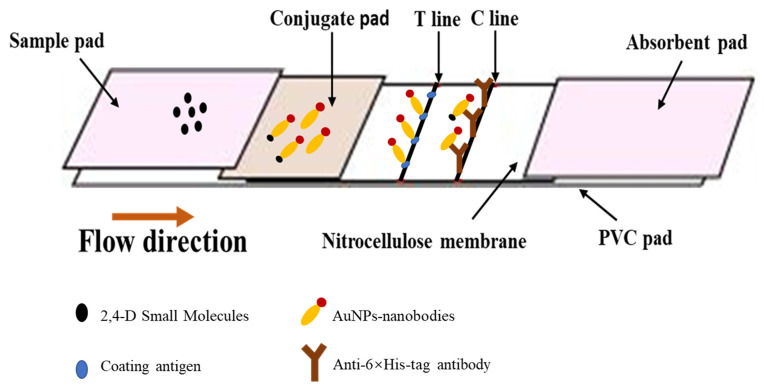
Assembly of test strips and principles of immunochromatography.

**Figure 2 biosensors-12-00084-f002:**
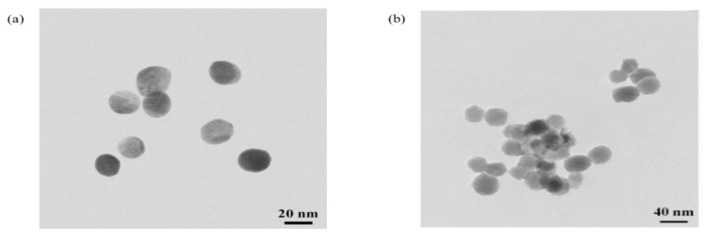
Transmission electron microscopy (TEM) images of the gold nanoparticles with a diameter of approximately 20 nm (**a**) and 40 nm (**b**).

**Figure 3 biosensors-12-00084-f003:**
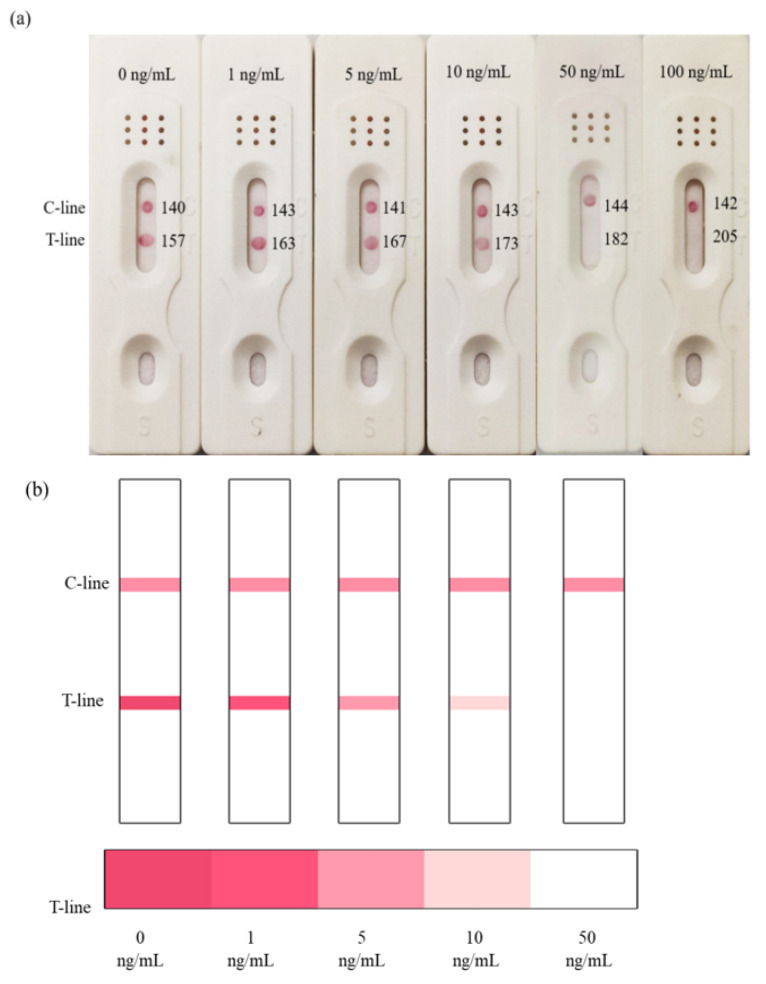
Determination of the visual detection limit of 2,4-D immunochromatographic test strip. (**a**) Different concentrations of 2,4-D were detected by the strips; (**b**) the simulated strips and “colorimetric card” based on the (**a**) results; (**c**) the calibration curve drawn based on the data of “Color Snap”.

**Figure 4 biosensors-12-00084-f004:**
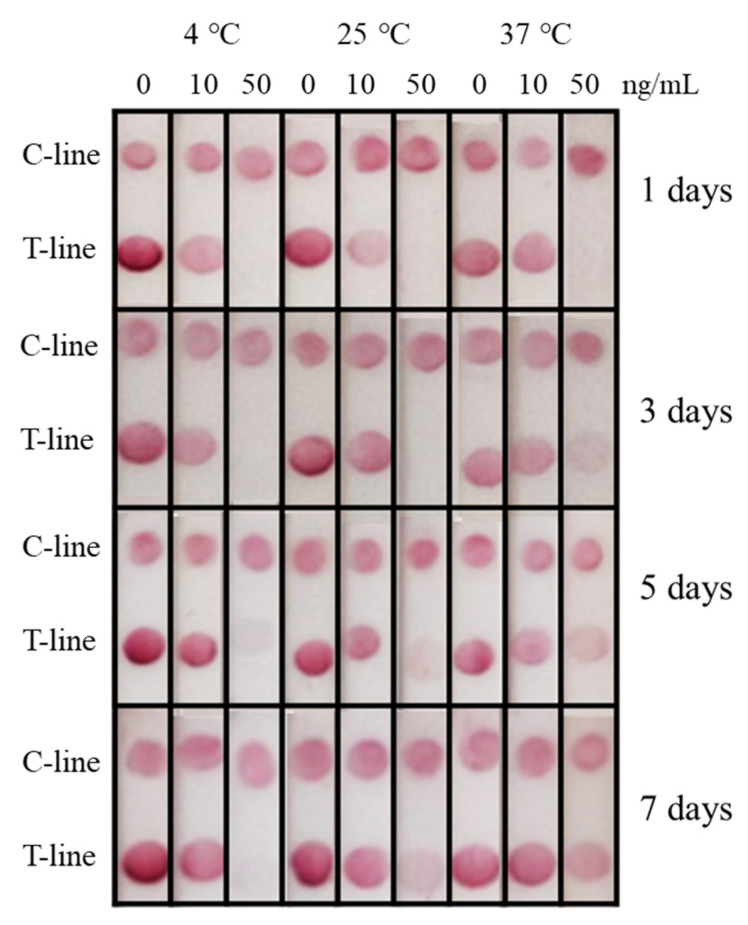
Stability of the immunochromatographic test strips for 1 days, 3 days, 5 days, 7 days at three temperatures (4 °C, 25 °C, 37 °C).

**Figure 5 biosensors-12-00084-f005:**
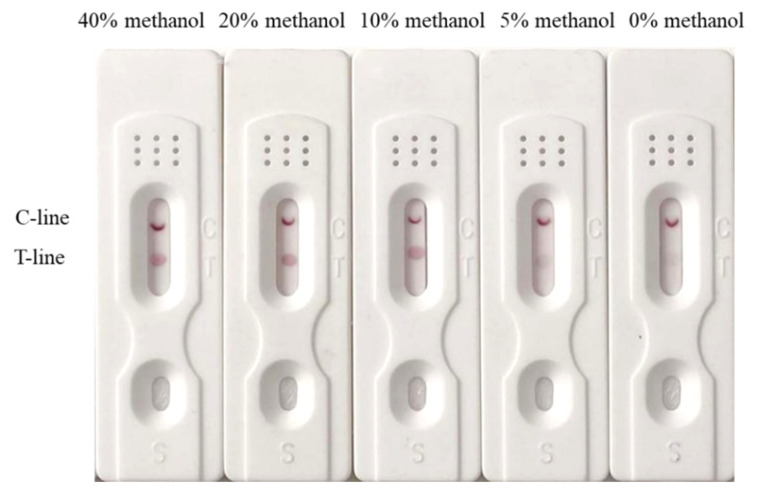
Evaluating the effect of methanol on the test line.

**Figure 6 biosensors-12-00084-f006:**
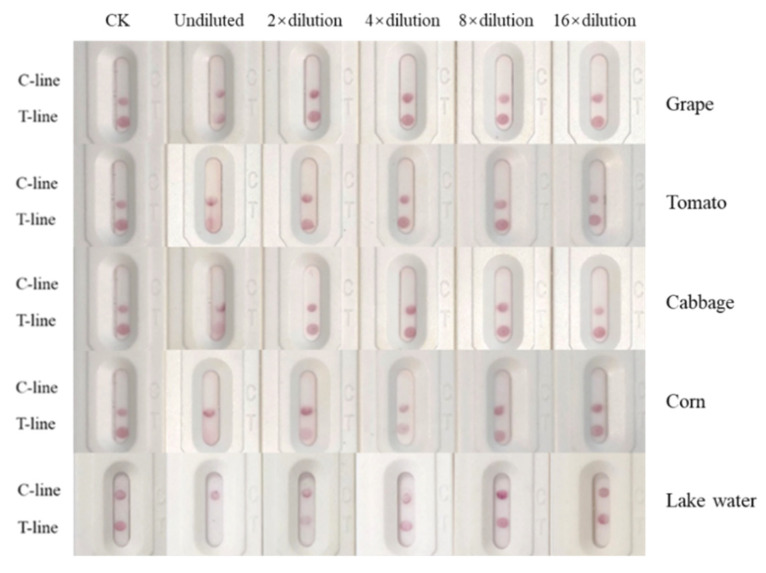
Effects of grape matrix, tomato matrix, cabbage matrix, corn matrix and river water matrix on test paper performance.

**Figure 7 biosensors-12-00084-f007:**
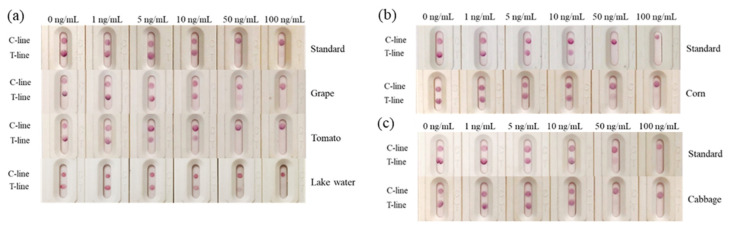
Spiked samples analysis; (**a**) the grapes, tomatoes and river water samples; (**b**) the cabbage samples; (**c**) the corn samples.

**Table 1 biosensors-12-00084-t001:** Cross-reactivity of the developed test strips against 2,4-D structural analogs.

Compound	Molecular Structures	Lateral-Flow Immunoassay	Cross Reactivity (%) ^1^ [21]
Negative control (0.01 M PBS (pH 7.4))	-	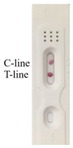	-
2,4-D	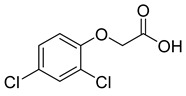	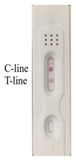	100
2,4-Dichlorophenoxy butyric acid	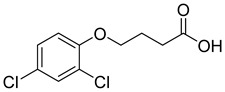	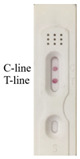	1.8
2-Methyl-4-chlorophenoxyacetic acid	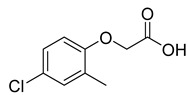	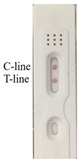	4.5
2-(4-Chloro-2-methylphenoxy)propanoic acid	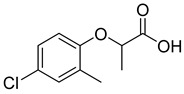	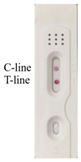	0.8
2,4-D methyl ester	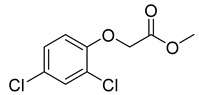	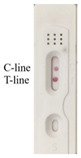	1.5

^1^ The cross-reactivity data referred to the previous research [21].

## Data Availability

The data presented in this study are available on request from the corresponding author.

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
