# Peer review of "Development of a Rapid Gold Nanoparticle Immunochromatographic Strip Based on the Nanobody for Detecting 2,4-DichloRophenoxyacetic Acid"

_biosensors, 2022, doi:10.3390/bios12020084_

Round 1

Reviewer 1 Report

This paper deals with the development of Au-NP-based immunochromatographic strips for detecting the 2,4-D herbicide in food/liquids. It has presented a “naked eye” visual detection limit of 50 ng/mL and a software assisted detection limit of 11 ng/mL. The cross-reactivity tests have shown good specificity and the stability tests showed that the strips are stable for long storage times at low temperatures (4ºC), whereas at room temperature (~25ºC) the strips become unstable after 5 days of storage.

The results are overall interesting, with enough data to backup all claims. They are clearly presented, although the English can be improved in several parts of the manuscript. I will address some minor points separately:

1 - I would suggest that some explanation is given to the observed variations in the absorbance values (starting on line 213, regarding Figure S-3).

2 - On the cross-reactivity part (section 3.5) the title is misspelled as a copy of section 3.4.

3 - In Figure S-4 and Figure 3c the ordinate label appears with a different text orientation as in Figure S-3. It’s best that a similar format is used across all images.

4 – Concerning English/grammar:

> It should be improved, particularly in lines 30-33, lines 244-245 and line 351.

> All acronyms should be clearly explained for the first time they are mentioned in the text (as e.g. is not happening in line 42).

> In line 149, “measured” is misspelled.

> In line 150, “adequate” is misspelled.

> In line 213 and line 230, “with the increasement” - shouldn’t be spelled instead “with the increase”?

> In lines 280-281, “which because that the structures” - shouldn’t be spelled instead “because the structures”?

> In line 282, “Cl atoms was replaced” - shouldn’t be spelled instead “Cl atoms were replaced”?

> In line 283-284, “which consistent with” - shouldn’t be spelled instead “which is consistent with”.

> In line 348, “the results were shown” - shouldn’t be spelled instead “the results have shown”?

> In line 376, “had the advantage of fast,” - shouldn’t be spelled instead “had the advantage of being fast,”?

Reviewer 2 Report

Junhao Chen et al. developed an immunochromatographic strip for rapid 2,4-Dichlorophenoxyacetic acid (2,4-D) detection. The manuscript is scientifically interesting, however, the novelty of such a test is not well explained. There are different analytical systems that detect 2,4-D more sensitively.

The main issues which need to be addressed and corrected in the manuscript:

  • It is very strange (unusual) that the limit of detection is higher than the limit of quantification. Is it a mistake or are the calculations not correct for such a format of immunoassay?
  • The authors are stating that it is a rapid test, however, the time of analysis is not mentioned in the Abstract.
  • Line 59 – should be enzyme-linked immunosorbent assay (ELISA) instead of enzyme-linked immunoassay (ELISA).
  • Lines 62-63 - According to cited references, ELISA should be instead of '... the immunoassay methods'.
  • In the text, “color snap APP” and “color snap App” are used. Please unify in the document.
  • 2. Construction of a nanobody-gold nanoparticle. Explain the principle of nanobody immobilization on gold nanoparticles. Is it simple adsorption? How the optimal concentration of nanobody was selected? The purification/centrifugation of nanoconjugate is not correctly described. g (force) or RCF (Relative Centrifugal Force) should be identified. How the concentration of obtained conjugates was calculated?
  • Additionally, why K2CO3 was added to the already synthesized solution of gold nanoparticles, and why does the absorbance at 520 nm have a tendency to increase? Why it is a good result?
  • What is the optimal pH for the nanobody? Is pH 9.5 good?
  • Figure 1 is not informative. The principle of immunoassay is unclear. Please show in the picture how each line was modified and how conjugate works in the analyte detection.
  • Line 104 – please provide information about the type of TEM and company.
  • Figure 3C – units of 2,4-D concentration should be added.
  • Figures 4,5,6,7 – too many pictures of immunochromatographic test strips. Could you present the obtained results as graphs or columns? It would be more informative for the readers.

Round 2

Reviewer 2 Report

The authors improved the manuscript, however, additional corrections are still required.

Authors are writing that the nanobodies with the size of about 15 kDa (Figure S-2) were used in this immunochromatographic strip test. However, a conventional antibody structure (yellow color) is presented in Figure 1. Additionally, why is the gold nanoparticle in the active center of antibody/nanobody? Is there only one site in the antibody/nanobody structure for the binding of nanoparticles?  I don’t understand why the authors made such a decision. Thus, the authors mislead readers.

Author Response

Point 1: Authors are writing that the nanobodies with the size of about 15 kDa (Figure S-2) were used in this immunochromatographic strip test. However, a conventional antibody structure (yellow color) is presented in Figure 1. Additionally, why is the gold nanoparticle in the active center of antibody/nanobody? Is there only one site in the antibody/nanobody structure for the binding of nanoparticles? I don’t understand why the authors made such a decision. Thus, the authors mislead readers.

Response 1: Acknowledging mistakes and having made changes in the pictures in the paper.